# Investigation to identify individual socioeconomic and health determinants of suicidal ideation using responses to a cross-sectional, community-based public health survey

Helen Mulholland [1], Jason C McIntyre,[2] Alina Haines-Delmont [3], Richard Whittington,[4] Terence Comerford,[5] Rhiannon Corcoran[1]

For numbered affiliations see end of article.

**Correspondence to**
Helen Mulholland;
helen.mulholland@liverpool.ac.uk

## ABSTRACT

**Objectives** To address a gap in knowledge by simultaneously assessing a broad spectrum of individual socioeconomic and potential health determinants of suicidal ideation (SI) using validated measures in a large UK representative community sample.

**Design** In this cross-sectional design, participants were recruited via random area probability sampling to participate in a comprehensive public health survey. The questionnaire examined demographic, health and socioeconomic factors. Logistic regression analysis was employed to identify predictors of SI.

**Setting** Community setting from high (n=20) and low (n=8) deprivation neighbourhoods across the North West of England, UK.

**Participants** 4319 people were recruited between August 2015 and January 2016. There were 809 participants from low-deprivation neighbourhoods and 3510 from high-deprivation neighbourhoods. The sample comprised 1854 (43%) men and 2465 (57%) women.

**Primary outcome measures** SI was the dependent variable which was assessed using item 9 of the Patient Health Questionnaire-9 instrument.

**Results** 454 (11%) participants reported having SI within the last 2 weeks. Model 1 (excluding mental health variables) identified younger age, black and minority ethnic (BME) background, lower housing quality and current smoker status as key predictors of SI. Higher self-esteem, empathy and neighbourhood belonging, alcohol abstinence and having arthritis were protective against SI. Model 2 (including mental health variables) found depression and having cancer as key health predictors for SI, while identifying as lesbian, gay, bisexual, transgender or queer (LGBTQ) and BME were significant demographic predictors. Alcohol abstinence, having arthritis and higher empathy levels were protective against SI.

**Conclusions** This study suggests that it could be useful to increase community support and sense of belonging using a public health approach for vulnerable groups (e.g. those with cancer) and peer support for people who identify as LGBTQ and/or BME. Also, interventions aimed at increasing empathic functioning may prove effective for reducing SI.

## Strengths and limitations of this study

► This study identified a number of novel protective factors associated with suicidal ideation including: neighbourhood belonging, level of self-reported empathy and abstaining from alcohol.

► Participants represented a large, non-clinical, community sample, which is relatively novel within suicidality literature.

► Generalisability of these findings is enhanced given the mixture of disadvantaged and less disadvantaged areas and random sampling of addresses as well as the application of a statistical adjustment for demographic variation in non-responses.

► This study utilised a 2-week measurement timeframe which is relatively novel within suicide research but in line with clinical risk management practices.

► The large overall sample size mitigates the limitation of assessing suicidal thoughts using a single-item measure.

## BACKGROUND

The eradication of suicide is a key national and global health policy.[1 2] Approximately 9% of people, across cultures, will experience suicidal thoughts (i.e. suicidal ideation (SI)) at some point in their lifetime.[3] Over a third of these people will plan their suicide, while over half of these people who plan will attempt suicide.[3] The personal impact of SI has been equated to suffering severe asthma or alcohol dependence.[4] Despite this significant disease burden, SI remains largely untreated with just 34%–42% of people receiving clinical or non-healthcare support.[5] The key reasons for this do not seem to relate to structural factors such as treatment availability. Rather they relate to a low perceived need for treatment by individuals and a preference for personal rather than formal management.[5] These

preferences may reflect the historical stigma associated with mental health broadly, and suicide specifically.[1 2]

However, the findings referred to above come from the WHO World Mental Health Surveys[6] which exclude the UK. Lifetime prevalence of SI within the UK population has been estimated to be more than double that of the cross-national prevalence rate.[7] Given the significant prevalence of SI and the apparent reticence of individuals to seek formal support, SI identification and clinical intervention strategies are imperative both nationally and globally.[1 2]

In support of these priorities, research has focused on understanding SI, including underlying risk and protective factors. Risk factors specific to SI identified in previous research include: female gender, parent psychopathology, childhood adversities, the presence of a diagnosed mental disorder and psychiatric comorbidity.[8] However, existing research has been criticised for a narrow focus on factors associated with individuals, while excluding societal and cultural factors, such as relative inequalities and relational matters.[9] Indeed, in their systematic review of reviews post 2007, McClatchey and colleagues[9] summarised risk factors for SI, suicidal behaviours (i.e. suicide attempts) and suicide completion to include mental ill-health, physical health (e.g. traumatic brain injury, type 1 diabetes mellitus), health behaviours (e.g. smoking, substance use (including alcohol)), biopsychosocial factors (e.g. parental suicide), experience of abuse, internet use, cyber bullying, lesbian, gay and bisexual sexuality, unemployment, 'elementary' occupations such as cleaners, agricultural workers, veterinary surgeons, military veterans and environmental factors (e.g. access to means). Of note, McClatchey et al's review did not elaborate separate risk factors between SI and suicidal behaviour.

Current 'ideation-to-action' theories of suicide also describe the complex interplay of biological, psychological, environmental and cultural factors that influence the inception of SI and the progression from SI to behaviour, such as the Integrated-Motivational-Volitional (IMV) Model of Suicide[10] and the Three Step Theory.[11] The IMV Model of Suicide[10] suggests that an interplay between background factors, such as personal disposition, deprivation, adversity and negative life events, can generate feelings of defeat and/or humiliation. These feelings endure and underpin perceptions of entrapment when negative appraisals of the personal agency and/or motivation to overcome such defeat/humiliation are experienced, leading to the development of SI. This inability to generate and implement positive solutions to personal problems may be due to the processes of cognitive restriction and deconstruction described by Baumeister[12] and/or emotional dysregulation[13] whereby individuals oscillate between emotional sensitivity/reactivity and emotional inhibition.[14] Alternatively, the social determinants of individuals' mental and physical health, rather than individual personal agency, may underpin perceptions of entrapment and lack of control.[15]

Although theory, evidence and policies suggest numerous personal and environmental risk and protective factors relating to SI, single studies that simultaneously assess a broad spectrum of individual socioeconomic and health determinants of SI using validated measures in large representative community samples are extremely scarce. In one such example, using the Composite International Diagnostic Interview, Handley et al[16] concluded that younger age, being unmarried, lifetime anxiety or lifetime post-traumatic stress independently predicted SI in an Australian rural community sample, after having controlled for lifetime depression.

Using responses to a community-based Household Health Survey (HHS), this study aimed to address this gap in the literature. Exploratory analyses sought to identify wider determinants of SI, with a specific focus on the impacts of health inequalities, to identify potential risk and protective factors specifically pertinent to SI experienced across a 2-week timeframe. Therefore, a subset of survey responses to demographic, socioeconomic, housing and neighbourhood, mental health, physical health, well-being, lifestyle and social capital domains were explored. The dataset thus allowed the examination of understudied phenomena in the SI literature, such as housing quality, caring responsibilities and medication side effects, which may underpin perceptions of entrapment. Indeed, poor housing quality—defined as accommodation with condensation, mould or fungus—has been shown to have a detrimental impact on both mental and physical well-being.[17] Carer burden has been identified as a significant risk factor, suggesting up to a fourfold increased risk of SI among carers across different patient populations, such as HIV,[18] chronic disease,[19] dementia[20] and cancer[21] compared with the general population. The literature regarding the link between medication side effects and SI is limited to clinical populations and antipsychotics/antidepressants. Current evidence suggests that 'Treatment-Emergent Suicidal Ideation' is relatively uncommon in older depressed adults.[22] One study exploring the psychiatric side effects of chloroquine and hydroxychloroquine found some evidence suggesting a weak positive association with SI.[23] Conversely, measures of empathy and social capital may act as protective motivational moderators between perceptions of entrapment and the development of SI, representing greater conformity to social norms/attitudes and perceived social belonging and/or support, also described in the IMV model of suicide.[10] Indeed, Zhang and colleagues[24] suggest that higher empathy could strengthen social deterrents of SI, thereby providing some support for this assertion. Further, a recent systematic review of reviews concluded that both objective social isolation and subjective perceptions of loneliness are risk factors for SI.[25]

This investigation sought to shed light on the role of relatively neglected determinants of SI, to examine if they predict SI over and above the effects of known risk factors, such as mental health problems, multimorbidity

and economic adversity, with a view to informing suicidality policy, prevention and risk management practice.

## METHODS

### Participants and sampling procedure

A cross-sectional HHS was conducted in the North West of England as part of the National Institute for Health Research Collaboration for Leadership in Applied Health Research and Care—North West Coast (NIHR CLAHRC-NWC). A random area probability sampling strategy was adopted. Twenty high-deprivation neighbourhoods and eight less-deprived neighbourhoods were selected, and random addresses were contacted within those neighbourhoods. The areas were selected in consultation with local authority representatives based on the following considerations: population size (5000–10 000 people), level of disadvantage (as measured via Index of Multiple Deprivation), coherent shared identity and available infrastructure for policy delivery. Overall, 4319 people were recruited between August 2015 and January 2016. There were 809 participants from low-deprivation neighbourhoods and 3510 from high-deprivation neighbourhoods. This was consistent with the sampling strategy, which had higher targets for high-deprivation areas due to the overall project focus of health inequalities. The sample comprised 1854 (43%) men and 2465 (57%) women with ages ranging from 18 to 95 years (M=49.12, SD=19.13). Consistent with the demographic composition of the region,[26] most participants (89%) indicated that they were from White European ethnic backgrounds. Participants were reimbursed with a £10 voucher in return for their participation. The adjusted response rate was 61%. A more detailed description of the sampling method and neighbourhood selection procedures can be found in Giebel et al.[26]

### Patient and public involvement

Five public advisors (PAs) from the National Institute for Health Research Applied Research Collaboration North West Coast (NIHR ARC NWC) were recruited as investigators based on their personal interest and/or experience of suicide and/or self-harm. PAs had an equal voice within the project team, which also comprised academics, researchers and clinicians. One PA agreed to be a named coauthor within the project dissemination materials, while the remaining four PAs declined named coauthorship. PAs helped to shape the research question, key objectives and variables of interest and contributed to the choice of statistical models used. The PA coauthor reviewed the paper commenting on accuracy and ensured the wording was accessible to the public. The PA coauthor was asked to prepare a plain English summary of the paper for inclusion on a university website, accessible to the public. This advisor has also agreed to be available for wider dissemination of the study results at conferences and with local interest groups as agreed with research personnel.

### Measures

A subset of the overall HHS questions was included in the analysis reported here. Decisions about which variables to include were informed by current SI theories and research evidence, as well as by extensive consultation with members of the project team, including clinicians, academics and people with lived experience. All variables were derived from single or multiple items of existing instruments recoded where necessary to between two and five categories for analysis. Coding and sources for all study measures are provided in online supplementary table 1. Information about **SI** was derived from responses to item 9 in the Patient Health Questionnaire (PHQ-9)[27] which elicits the frequency of 'thoughts that you would be better off dead or of hurting yourself' in the preceding 2 weeks. Responses of 'several days' or higher frequency were coded as '1' and 'not at all' as '0'. **Sociodemographic variables and caring responsibilities** were coded in accordance with UK Office for National Statistics national census categories.[28] Other variables were measured as follows: **housing quality**—English Housing Survey,[29] 3 items; **financial situation**—Wealth and Assets Survey,[30] 1 item; **physical health status**—EQ-5D,[31] 5 items; **social capital and neighbourhood belonging**—Community Life Survey,[32] 3 items; **physical health conditions**—Adult Psychiatric Morbidity Study,[33] 23 conditions, 1 item each; **medication side effects**—Health Survey for England,[34] 2 items; **alcohol consumption and smoking**—Merseyside Lifestyle Survey,[35] 1 item each; **depression**—PHQ-9,[27] 8 items as item 9 (SI) was used as the dependent variable; **anxiety**—Generalised Anxiety Disorder Questionnaire (GAD7),[36] sum of 7 items; **paranoia**—Five-item Persecution and Deservedness Scale (PaDS-5),[37] sum of 5 items; **well-being**—Warwick-Edinburgh Mental Wellbeing Scale (WEMWBS),[38] sum of 7 items, abbreviated; **self-esteem**—Self-Esteem Scale,[39] 1 item; **empathy**: Interpersonal Reactivity Index (IRI),[40] sum of 5 items, abbreviated; **hopelessness**—sum of 2 items (*Brief-H-Pos:* reverse scored)[41]; and **locus of control**—Levenson Locus of Control Scale,[42] sum of 9 items, abbreviated. Descriptive statistics for each measure are provided in online supplementary table 2.

### Data analysis strategy and preliminary analyses

Data were analysed using Stata V.12.[43] As the dependent variable, SI, was highly skewed (S-W=0.92, p<0.00001), the variable was recoded into 0=*suicidal ideation absent*, 1=*suicidal ideation present*. While dichotomisation of variables can result in potential reductions in effect sizes and power, as well as loss of information, it is recommended for instances of severely skewed data where many participants fall at the extreme end of a scale as is the case here.[44] Specifically, 89% (n=3833) of the sample reported having no SI over the previous 2 weeks, while 454 participants reported having SI (every day: n=99; more than half the days: n=138; several days: n=217). Given the possibility of collinearity between the four mental health symptoms and between mental health symptoms and SI, Pearson's

product moment and Pearson's point-biserial correlations were conducted to examine bivariate relationships. As shown in table 1, all predictors were moderately correlated with the criterion. The strongest association was between depression and SI, $r_{pb}$ (4285)=0.57, p<0.001. When examining collinearity between predictors, anxiety and depression were highly significantly positively correlated, r (4303)=0.79, p<0.001. As the correlation was below 0.8 and anxiety and depression represent distinct theoretical constructs, multicollinearity was not considered problematic for the logistic regression (LR) analysis.[45]

### LR Analyses without adjusting and adjusting for mental health variables

Two LR analyses were conducted with SI regressed on the socioeconomic, health and lifestyle variables. Standard errors were adjusted to account for the clustered nature of the data using the *svyset* command and the 28 neighbourhoods as clusters. The data were also weight-adjusted to account for demographic variation in non-response. The models provided estimates of the OR of SI associated with each variable, while holding all other variables in the model constant. Because the mental health symptoms explain a substantial portion of variance in SI, we constructed models both excluding (Model 1) and including (Model 2) symptoms to quantify the association between social and health factors and SI, as well as their predictive power above and beyond the effects of mental health.

Analysis indicated that no variable was missing more than 5% of values and only one variable (housing quality) was missing more than 1% of values. Little's Missing Completely At Random (MCAR) test indicated data were not missing completely at random, $\chi^2$(335)=457.35, p<0.001. Follow-up separate variance t-tests with threshold set to 1% indicated that housing quality missingness was associated with the mental health indicators of depression, anxiety, paranoia and well-being (ps<0.005). Because Little's MCAR is highly sensitive to large sample sizes and missingness was extremely low for all variables, listwise deletion was used to account for missing values in each analysis. This resulted in n=3966 for Model 1 and n=3940 for Model 2.

**Table 1** Bivariate correlations between mental health variables

| Variable | 1 | 2 | 3 | 4 | 5 |
|---|---|---|---|---|---|
| 1. SI | – | 0.57* | 0.51* | 0.34* | −0.34* |
| 2. Depression | – | – | 0.79* | 0.50* | −0.52* |
| 3. Anxiety | – | – | – | 0.55* | −0.52* |
| 4. Paranoia | – | – | – | – | −0.39* |
| 5. Well-being | – | – | – | – | – |

*p<0.001.
SI, suicidal ideation.

## RESULTS

### Model 1: LR predicting SI without adjusting for mental health variables

The overall model was significant, $F$(62,3877)=6.42, p<0.0001. Significant effects with alpha set to 0.001, 0.01 and 0.05 are highlighted and adjusted ORs are reported alongside confidence intervals within table 2, while both significant and non-significant effects for all variables are reported in the online supplementary table 3. Age was a significant predictor of SI. All younger age groups reported significantly higher odds of SI compared with the base category of 65+ years. Eighteen to 24-year olds had the highest increase in odds of SI relative to the base category. People from black and minority ethnic (BME) backgrounds had significantly higher odds of SI compared with people from White European backgrounds. Living in lower quality housing, being in the same financial position as 12 months ago and not currently being employed were all significantly associated with higher odds of SI.

Experiencing moderate or extreme pain/discomfort increased the odds of SI. Having side effects from medication was associated with higher odds of SI. Of the physical health condition variables, having a stroke or a hearing condition in the previous 12 months was associated with significantly increased odds of SI. Reporting arthritis was associated with significantly lower odds of SI. Examination of the psychological risk factors of mental illness revealed that higher levels of self-esteem were significantly associated with lower odds of SI. Similarly, higher levels of empathy were associated with lower odds of experiencing SI. Conversely, higher levels reported on the external locus of control 'chance' subscale were significantly associated with higher odds of SI. Feeling hopeless was also associated with higher odds of SI. Lifestyle factors were also significantly associated with SI. Being a current occasional or heavy smoker was associated with higher odds of SI. Abstaining from alcohol reduced odds of SI by 37% relative to drinking within the recommended limits.[46] Of the social capital variables, neighbourhood belonging was the only significant predictor, whereby an increase in sense of belonging was associated with lower odds of SI.

### Model 2: LR predicting SI adjusting for mental health variables

The overall model was significant, $F$(69,3844)=9.38, p<0.0001; however, the profile of significant risk factors was somewhat different compared with Model 1, as reported in table 2. After adjusting for mental health symptoms, identifying as lesbian, gay, bisexual, transgender or queer (LGBTQ) or BME was associated with significantly higher odds of SI. Reporting being in the same financial position as in the previous 12 months was significantly associated with 2.29 times higher odds of SI compared with being in a worse position than 12 months ago. No other demographic or socioeconomic variables significantly predicted SI.

Reporting a cancer diagnosis was significantly associated with 3.90 higher odds of SI, while reporting arthritis

**Table 2** Statistically significant logistic regression variables predicting SI excluding (Model 1) and including (Model 2) mental health variables

| | Model 1 | | Model 2 | |
|---|---|---|---|---|
| Predictors | Adjusted OR of SI | 95% CI | Adjusted OR of SI | 95% CI |
| **Mental health** | | | | |
| Depression | – | – | 7.24*** | 5.22–10.07 |
| Anxiety | – | – | 1.56** | 1.13–2.17 |
| Paranoia | – | – | 1.36* | 1.07–1.72 |
| **Demographics** | | | | |
| Age (65+) | | | | |
| 18–24 | 5.50*** | 2.74–11.06 | 0.95 | 0.38–2.38 |
| 25–44 | 4.50*** | 2.48–8.15 | 1.62 | 0.84–3.15 |
| 45–64 | 2.82** | 1.68–4.73 | 1.10 | 0.60–2.02 |
| BME | 1.88* | 1.01–3.49 | 1.93* | 1.04–3.62 |
| LGBTQ | 1.93 | 0.77–4.83 | 2.73* | 1.00–7.46 |
| **Socioeconomic status** | | | | |
| Problems with housing | 1.67*** | 1.26–2.23 | 1.34 | 0.95–1.89 |
| Financial position (worse) | | | | |
| Same | 1.68* | 1.02–2.76 | 2.29** | 1.24–4.23 |
| Non-employment | 1.43* | 1.00–2.03 | 1.06 | 0.68–1.65 |
| **Health problems** | | | | |
| Pain | 1.62* | 1.09–2.40 | 0.98 | 0.61–1.56 |
| **Side effects** | | | | |
| No medication | | | | |
| Never bother | 1.47* | 1.02–2.12 | 1.25 | 0.77–2.03 |
| Bother a little | 2.93** | 1.35–6.36 | 1.72 | 0.64–4.67 |
| Bother somewhat | 2.31** | 1.23–4.34 | 0.83 | 0.37–1.86 |
| Bother a lot | 2.64** | 1.37–5.10 | 0.72 | 0.25–2.03 |
| **Health conditions** | | | | |
| Cancer | 1.74 | 0.80–3.77 | 3.90** | 1.40–10.84 |
| Ear | 2.02** | 1.20–3.41 | 1.24 | 0.59–2.59 |
| Stroke | 2.01* | 1.06–3.81 | 1.63 | 0.57–4.68 |
| Arthritis | 0.59* | 0.40–0.88 | 0.54* | 0.30–0.95 |
| **Alcohol consumption** | | | | |
| None (0 units) | 0.63** | 0.46–0.87 | 0.61* | 0.42–0.90 |
| **Smoking status** | | | | |
| Current occasional smoking | 1.99* | 1.04–3.81 | 1.78 | 0.80–3.96 |
| Current daily smoking | 1.92*** | 1.35–2.74 | 1.51 | 0.98–2.33 |
| **Psychological factors** | | | | |
| Empathy | 0.82* | 0.70–0.96 | 0.72** | 0.59–0.88 |
| Self-esteem | 0.81*** | 0.75–0.88 | 0.97 | 0.87–1.09 |
| Locus of control (chance) | 1.35** | 1.11–1.64 | 1.23 | 0.95–1.60 |
| **Social capital** | | | | |
| Neighbourhood belonging | 0.69* | 0.48–0.97 | 0.90 | 0.58–1.38 |

*p<0.05, **p<0.01, ***p<0.001.
BME, black and minority ethnic; LGBTQ, lesbian, gay, bisexual, transgender or queer; SI, suicidal ideation.

was associated with reduced odds of SI. Self-esteem, hopelessness and locus of control were not associated with SI when mental health variables were taken into account. However, increased empathy scores were associated with a reduction in odds of SI. Past and present smoking behaviours were unrelated to SI in this model. Abstaining from alcohol was significantly associated with lower odds of SI. No social capital variables were associated with SI. All of the mental health symptom variables were associated with higher risk of SI. Specifically, anxiety and paranoia were associated with significantly higher odds of SI while depression showed the strongest relationship with SI, with each 1 unit increase on the PHQ-9 being associated with 7.24 higher odds of SI. Well-being was not related to SI.

## DISCUSSION
### Principal findings
This study was uniquely able to investigate wider determinants of SI including demographic, socioeconomic, housing and neighbourhood quality, mental health, physical health, well-being, lifestyle and social capital factors. Utilisation of a community-based population enhances generalisability of these findings beyond the clinical populations typically used within suicidology literature. In addition, the application of a shorter measurement timeframe (i.e. 2 weeks) within suicide research is relatively new, but in line with clinical risk management practices.

Congruent with current suicide prevention literature[47] depression, anxiety and paranoia were all identified as risk factors for SI. The strongest of these effects was related to depression insofar as each 1 unit increase on the PHQ-9 was associated with a sevenfold increase in odds of SI.

Physical health conditions that are enduring and/or debilitating in nature, or life threatening, have been shown to correlate with SI both dependently[48] and independently from common mental disorders[49] and our study supports these findings. Specifically, pain/discomfort, having cancer, a stroke or hearing problems may engender perceptions of burdensomeness,[50] defeat and entrapment,[51] psychological and/or physiological pain and/or hopelessness,[52] which are all suggested preconditions for SI. However, 'arthritis' was found to be a protective factor against SI in this sample. A possible explanation could be that this is a common condition, particularly among older people and therefore individuals may feel less 'alone' living with arthritis and/or there may be less stigma and more formal/informal support for sufferers. Another potential explanation of this unexpected finding is that treatments for arthritic pain may have antidepressant effects.[53 54] This finding requires further investigation.

Perceptions of defeat and/or entrapment may also underpin findings from previous research which have shown that family caregivers of patients with cancer and dementia have higher levels of SI with comorbid depression, while older age and having clear reasons for living reduce such risks.[55] However, carer status was not found to be a statistically significant predictor in this sample. Carer burden may be exacerbated by longer duration of carer role and particularly challenging patient needs. These aspects of caring were not examined within this survey and may explain the differences in findings.

Poor quality housing, being in the same financial position as the previous year and being unemployed were identified risk factors for SI within this study, which reflect the known wider determinants of health inequalities[56] and may also represent perceptions of defeat/entrapment that underpin SI. Without adjustment for mental health factors, younger age—particularly being aged between 18 and 24—was found to increase SI risk, which is again in line with previous literature.[8] Similarly, hopelessness and believing your life to be determined by 'chance' were also risk factors for SI. These findings may reflect reduced objective or subjective personal agency and an opportunity for targeted educational, occupational and clinical interventions.

Our findings show that higher self-esteem is a protective factor against SI, again corroborating previous research.[57 58] Further, both theory and research suggest that higher levels of social capital have a positive impact on mental health[59] and our analysis excluding mental health factors supports this.

Thwarted belonging relates to the fundamental need to belong and when this need is compromised it can underpin SI and behaviour.[50] Indeed, Wastler *et al*[60] found that in a sample of people with first episode psychosis, perceived burdensome and thwarted belonging were elevated in people with recent SI compared with individuals without recent SI. Joiner's[50] theory may explain the increased risk of SI from minority status groups such as identifying as LGBTQ or being from an ethnic minority background, as demonstrated within previous research[61] and by the results from this study. Further, as the sample was from a predominantly White British population, it is possible that social factors such as discrimination and social exclusion experienced by people from BME groups may have contributed to their greater SI vulnerability. In support of this, neighbourhood belonging was found to be a protective factor against SI within this study, providing additional support to the argument that a sense of belonging can support better mental health. Importantly, neighbourhood belonging has been found to be more prominent in lower socioeconomic status (SES) populations than in higher SES populations where wider social networks play a more prominent role in sense of belonging.[62]

Given the reported preference for self-management of SI,[5] individuals may use lifestyle behaviours such as

smoking and alcohol consumption as coping mechanisms. Our findings highlight smoking and higher levels of alcohol consumption as risk factors for SI.

Our findings indicate that higher levels of empathy reduce the risk of SI which remain when adjusting for mental health factors. Zhang and colleagues'[24] suggestion that higher empathy strengthens social deterrents to SI and behaviour provides a possible explanation for this novel finding in a non-clinical sample. Having the automatic capacity to take the perspective of loved ones left behind when one is contemplating suicide would provide a strong, natural barrier to end such thoughts and to bar completion.

## Limitations

While based on validated measures with a large representative sample, the methodology adopted here has certain limitations which must be considered when interpreting the results. First, the survey used in this study is entirely based on self-report methods. Reporting bias can be an issue, due to the sensitive nature of some of the interview questions. Second, capturing SI using a single-item measure may have resulted in over simplification, as the validity of single items when detached from a larger instrument can be contested. However, the large sample size may have mitigated these problems. Third, again despite the large sample size, generalisability beyond the sample studied here must be exercised with caution. This was a regional rather than a national sample and there may be specific economic or cultural factors which do not apply beyond the North West of England. The demographic characteristics of the sample were restricted in terms of age and ethnicity. The exclusion of participants aged under 18 years prevents conclusions being drawn about the adolescent age group where SI is common. Similarly, the aggregation of non-White ethnicities into a single BME category in the analysis here eradicates any possible examination of differences within these non-White groups. Fourth, the SI outcome variable does not provide information on actual suicidal behaviour which is the key clinical need to be addressed. Finally, the SI variable was also very skewed and the variable needed to be dichotomised to increase power and minimise error variance. However, this was at the expense of nuance in the findings insofar as our data cannot elucidate potential differences between higher and lower frequency SI.

## Clinical implications

While being mindful of the limitations of this study, some of the novel findings reported, if replicated, have clear clinical implications. Perhaps the most important of these is consistent with public mental health approaches to intervention including social prescribing routes for prevention of distress and promotion of well-being. These approaches stress the importance of building community and sense of belonging. Our findings indicate that peer support groups for chronic health conditions such as cancer, stroke and hearing could mitigate thoughts of suicide by providing social support, a reason to continue and a source of relational well-being, as well as potentially adding to knowledge about one's condition and how to cope with it. The same is true for particular groups shown in our analysis to be more prone to SI, including LGBTQ and ethnic minority groups. Neighbourhood support groups could go some way to increase a sense of belonging to a community for minority groups. Relational approaches to support individual mental health have been advocated for some time and there is a wealth of robust evidence demonstrating the role of peer support in enhancing well-being in the context of mental and physical heath difficulties.[63–66] Our analysis shows that the benefits of these approaches may well extend to the prevention of SI. Furthermore, it is likely that increased interpersonal contact with similar individuals in the context of support or neighbourhood groups may, in time, translate to enhanced perspective-taking skills which, in this sample, was found to be a psychological variable negatively associated with proneness to suicidal thoughts. Thus, communities of place, support and interest may provide solutions to the experience of SI and may prevent such thoughts escalating to suicidal acts.

## CONCLUSION

Identification of risk and protective factors for SI can support the implementation of tailored clinical and non-clinical interventions. This study has identified new risk and protective factors for SI using a randomly selected large community-based sample from disadvantaged and less disadvantaged neighbourhoods. Using this approach, as well as statistically adjusting for demographic variation in non-responses, enhances the validity of the study, especially the generalisability of its findings beyond the clinical populations typically used within suicidality literature. This study suggests that it could be useful to increase community belonging and community support within a public health approach for vulnerable groups (e.g. those with cancer) and peer support for people who identify as LGBTQ and/or BME. Also increasing empathic functioning, potentially through involvement with support groups may be an effective strategy for reducing SI.

**Author affiliations**
[1]Primary Care and Mental Health, University of Liverpool, Liverpool, UK
[2]Natural Sciences and Psychology, Liverpool John Moores University, Liverpool, UK
[3]Department of Nursing, Manchester Metropolitan University, Manchester, UK
[4]Brøset Centre for Research and Education in Forensic Psychiatry, St. Olav's Hospital and Department of Mental Health, Norwegian University of Science and Technology, Trondheim, Norway
[5]National Institute for Health Research Applied Research Collaboration North West Coast (NIHR ARC NWC), University of Liverpool, Liverpool, UK

**Acknowledgements** We would like to acknowledge Dr Cecil Kullu, Emma Mullin, Paula Gross, Jane Shelton, Stuart Wood and Farheen Yameen and posthumously acknowledge Elizabeth Fuller for contributing to the design of this project and its governance.

**Contributors** The authors have all substantially contributed to this paper. HM is the lead author who conceived the study, provided interpretation of the data for this paper and drafted all sections of the manuscript, with the exception of the Methods and 'Clinical implications' section. JCM has contributed to the design of the study, conducted the data analysis, drafted the Methods section and provided substantial critical comments to the whole manuscript at the drafting stage. AH-D has contributed to the design of the study and interpretation of the data, provided substantial critical comments at the drafting stage and conducted final manuscript editing and proofing. RW has contributed to the drafting of the Methods section, interpretation of the data and provided critical comments at the drafting stage. TC has contributed to the design of the study, interpretation of the data, drafted the 'Public and Patient Involvement' section and provided critical comments at the drafting stage. RC has contributed to the design of the study, interpretation of the data, drafted the 'Clinical implications' section and provided critical comments at the drafting stage. All authors have approved the final version of the manuscript.

**Funding** This study is part-funded by the NIHR CLAHRC NWC. TC is part funded/supported by The National Institute for Health Research Applied Research Collaboration North West Coast (NIHR ARC NWC).

**Competing interests** JCM reports grants from NIHR, during the conduct of the study.

**Patient consent for publication** Not required.

**Ethics approval** The research was approved by University of Liverpool Committee on Research Ethics (Ref: RETH00836 and IPHS-1516-SMC-192) and conforms to the principles embodied in the Declaration of Helsinki. Written informed consent was obtained from all participants.

**Provenance and peer review** Not commissioned; externally peer reviewed.

**Data availability statement** Data are available upon reasonable request. The dataset from the North West Coast Household Health Survey will be made publicly available after an embargo period.

**ORCID iDs**
Helen Mulholland http://orcid.org/0000-0002-6679-7257
Alina Haines-Delmont http://orcid.org/0000-0001-6989-0943

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
