## [Reviewer comments · BMJ Open]

ARTICLE DETAILS

TITLE (PROVISIONAL)	An investigation to identify individual socio-economic and health determinants of suicidal ideation using responses to a cross-sectional, community-based public health survey.
AUTHORS	Mulholland, Helen; McIntyre, Jason; Haines-Delmont, Alina; Whittington, Richard; Comerford, Terence; Corcoran, Rhiannon

VERSION 1 – REVIEW

REVIEWER	Hyoun S. Kim University of Calgary
REVIEW RETURNED	06-Nov-2019

GENERAL COMMENTS	Thank you for the opportunity to review the present manuscript exploring predictors of suicidal ideation and self-harm in a large representative sample of adults. The study had several noticeable strengths including a representative sampling design and use of validated measures. I provide my comments below in hopes of clarifying and strengthening the present manuscript. 1. Abstract: The authors note that model 1 excluded mental health variables, yet a significant predictor is comorbid physical and mental ill-health. I presume this is in reference to the variable "physical health conditions" which includes coding for mental health conditions. I would remove this variable for model 1 and include only in model 2 for clarity.2. Abstract and Methods: The authors group alcohol and smoking as lifestyle factors. SUDs could also be classified as mental health disorders. The authors should provide a rationale for including alcohol and smoking as lifestyle factors rather than as a mental health variable and or include these variables only in Model 2 as mental health variables.3. Background: It would be helpful if the authors provide more context and background regarding the novel determinants of SI. Specifically, what is the rationale for exploring housing quality, caring responsibility, specific health conditions as predictors of SI?4. Methods: For the creation of the SI variable, I assume the authors coded from the PHQ-9, 0 as being absent of SI and 1-3 indicating presence of SI? It would be helpful if the authors provide frequencies of people who scored 0,1,2,3 on the item 9 for detailed reporting purposes.5. Methods: Related to the above comment, if there is an adequate sample size, conducting a multinomial logistic regression (0= No SI, 1 = several days, more than half the days and 2 = nearly everyday) may be highly informative. People who experience SIs nearly everyday are qualitatively different than those who experience SIs several days or more than half the days and may have unique risk and protective factors.
--

	6. Methods: One important question I have is the calculation of the PHQ-Score. Given the authors used item 9 as their dependent measure, was this taken into account when calculating participants total score on the PHQ-9? If not, this may violate assumptions of independence and also contribute to the strong association between depression and SI. 7. Method: Were the measures reported herein all measures included in the survey? If not, the authors should explicitly state so and provide a rationale for selectively choosing their measures, especially in light of their aim of identifying novel predictors. 8. Method: Please provide results of Little's MCAR. If the data is missing at random, it would help justify the authors choice of using list wise deletion. 9. Method: I am curious from an ethics and safety perspective, if any safety checks were in place for high-risk participants, specifically during who reported experiencing SIs nearly everyday. 10. Patient and Public Involvement: I would suggest moving the first paragraph of this section to the Methods, and incorporating the second paragraph in the discussion to help with the structure and flow of the manuscript. 11. Discussion: I believe the manuscript would be strengthened by a more detailed implication section in the discussion section. More specifically, in the abstract, the authors state that interventions should focus on reduction depression and enhancing self-esteem, social, capital and empathy. How exactly could this be done, especially in a community based context?
--	--

REVIEWER	Dr C Clements University of Manchester, UK
REVIEW RETURNED	05-Jul-2020

GENERAL COMMENTS	Interesting paper with some important and novel results of associations between social, economic, and lifestyle factors and SI in a community sample. It is a shame there was no measure of acts of suicidal behaviour as the ideation to behaviour conversion is typically low, and therefore it is difficult to say how relevant the result may be in clinical terms or for suicide prevention efforts. The methods and analysis are appropriate, but there are some concerns around the way the content is presented as it lacks a clear focus, with some details of the study not entirely clear until well into the paper and/or a reliance on supplementary materials. Therefore, while this paper is a good candidate for publication I would recommend a thorough review, with additional clarifications, details, and a clearer focus on what exactly is novel about this work, what gap in the current literature it is addressing, and how the results might be relevant clinically/practically. Simplification of the writing would also reduce word-count and allow for some things that have been shunted to Supplementary files to be brought back into the main text. The core of the paper is however solid, and I hope the following comments will be of use to the authors. Detailed comments on each section are provided below: Abstract. Background. The background section within the abstract doesn't really set up the reason why this work is necessary or what questions it answers. The novelty of the study is not important
---

here, instead, the aim of the work should be stated more clearly - what are you trying to address?

Methods. I would like to know a bit more about how suicidal thoughts were assessed within the survey. What type of measures/response options were included. If this study is a sub analysis of a larger study, please include the name of the study and clarify the situation e.g. that the data were primarily collected for another purpose. What sort of regression analysis? Multivariable? What was controlled for?

Results. What does additional statistically significant predictors were noted mean? If they weren't relevant to your conclusions they don't need to be mentioned in the results of the abstract. The model 1 and 2 (or adjusted and unadjusted models) should be referred to in the methods if necessary rather than here.

Conclusion. The conclusion is a bit abrupt and could have a more reader friendly sentence summarising the implications of the main results before going on to suggest interventions or how the results could be useful in further research or practice.

Strengths and limitations.

Point one should be rephrased to focus on the novelty as a strength rather than just stating that it is novel. What exactly is the novel aspect - the comment is too broad to be clear, and certainly many of the variables are things that are well known within the literature. It would be better for the paper to draw out exactly what aspects are novel right from the start.

Point five is not clear without further details, furthermore, while the overall sample is fairly large, the number with SI is not so large, so this could be a bit misleading.

Background

General comments. Each statement should be supported by a reference. The background/introduction doesn't clearly demonstrate what this work is addressing that hasn't been addressed previously. The background needs to lead the reader to the conclusion that this work is necessary to address a gap in the current literature. I would caution against grouping suicidal ideation with suicide attempts and complete suicide in previous research as the ideation to action pathway is not that strong e.g. while most people who attempt or die by suicide will have had suicidal ideation the majority of people with suicidal ideation never go on to act on these thoughts - see for example work on the ideation-to-action framework e.g. Klonsky & May 2014, 2016 etc. More focus on previous work specifically on suicidal ideation, what remains unknown and how this work addresses it would improve the paper.

Methods

General comments. As there has been work published already from this survey that has details of the methods, you could name the study up front and refer the reader to the paper or report for further details of the main study to reduce the amount you have to cover here. Then you can focus on the SI relevant parts.

Measures. More details are needed in the measures section of the main text, particularly around the measures that were used to assess suicidal ideation. Rather than just adding supplementary

files it would be good to have some kind of summary in the actual text and refer to the supplementary files for more detailed information on the other variables (though I'm not sure the level of detail in the supplementary files is needed either - especially if you can refer back to an existing publication or the the original references for the measures). Also, I couldn't see the suicide ideation variable in Table S1 but as the item is scores on a 4-point scale it would be appropriate to include how it was reduced to a Y/N variable. I'm sure you could combine the study measures S file and Table S1 somehow to reduce the amount of additional files.

Data analysis. The issue of broad dichotomisation should be picked up in the limitations section - what might be the impact of this on the results? Small point - tables usually go at the end of the manuscript at submission rather than in the text.

Results

General comments. You could put some information about the sample demographics here instead of in the methods section. I'd like to know if there were any demographic differences in the SI and non-SI groups with the n% rather than just the ORs - numbers may be small in some analyses, but it is impossible to judge this at the moment. The tables need reviewing and simplifying - I can see why you put a shorter version in the text but this is missing information about the different levels or categories of the measures. Perhaps it would be better to include fewer variables, but in more detail (e.g. what variables are relevant, rather than just all that are significant), and remove these from the supplementary table to reduce duplication.

Model 2. Don't need to repeat what was adjusted for this should be covered in methods.

PPI

A bit more detail on how exactly they contributed to the study would be good. Did their influence change any questions or procedures, how did they help interpret the findings? Is this specific to this study or are you talking about the main survey work? Page 11 line 6 - I find this sentence about the authorship a bit strange. It implies more than one person had input into the manuscript but only one person was added as a co-author. Please define further why the contribution of one person, and not the rest of the public advisors, was considered significant enough to warrant authorship. If these additional details are provided, the paragraph starting line 11 (Page 11) is not needed.

Discussion

General comments. You need to identify what specifically is the strong point of this work and emphasise that rather than just repeating that a lot of variables were included. If the info around housing, lifestyle and wellbeing is novel - concentrate on that. The community sample is also a bonus as a lot of work in suicidology is clinical in nature. The principal findings are a bit long. Rather than just repeating what is significant as in your results section, you could focus on the most relevant or novel results. The structure of the discussion could be tightened up to make it easier for the reader to grasp the main points of the work. There is a lack of interpretation and translation of the results and an overemphasis on whether the results are consistent with previous research. This take up a lot of the text and as these are not new information could

	instead be summed up with a few simple sentences e.g. Many of the results were consistent with previous research on predictors of SI, such as X Y and Z (with references). This would free up space to focus on the novel results and include further interpretation. See detailed comments below: Page 11, line 40-46. This sentence is not a result and is repeating what we already know, would recommend removing. Page 11, paragraph starting line 49. This largely re-states what we have already been told in the results section. It would be better to concentrate on those results that have some kind of clinical or theoretical relevance - what results are novel, and how should these be interpreted? Please be careful of extraneous words E.g. the sentence “Unsurprisingly, depression was the most important risk factor, whereby those suffering from depression were seven times more likely to report SI than those without” could just read something like “Those suffering from depression were seven times more likely to report SI than those without.” In fact this could be combined with the later sentence (Page 12, line 29-31) about depression, anxiety and paranoia being known factors in current literature (a general writing tip - see how many words you can remove form a sentence and still have it make sense - I guarantee it is less than you think!). Limitations. The limitations section is seriously lacking. Issues that could be covered - geographical location of the survey and how this impacts generalisability to other areas. Only adults were included - self-harm is common in young people, what about SI? No acts of suicidal behaviour were recorded, and there was no follow-up to establish if thoughts were acted on - could these people be the ones who never act on these thoughts - if so, how useful is this data? The issues around collapsing categories could be raised here e.g. BAME people were all placed in the same group, but there might be nuances and cultural aspects that this hides. What about the 1-4 levels of SI? The reader does not get to see these at all (actually that is something you could include in the results - just n% of the SI group) and it could be that these are all low level thoughts. Conclusion The conclusion starts strong, with some interpretation that I would like to see more of in the rest of the discussion. But it ends with an overstating of the relevance of this work - see comments on additional limitations to consider. More detail needs to be added as to how this work can inform future research, clinical management, and reduction and prevention strategies. Again you could focus on the more novel aspects of your work. What about public health initiatives to tackle community SI? BAME or LGBT specific interventions?
--	--

VERSION 1 – AUTHOR RESPONSE

Reviewer 1

Reviewer Comment: Thank you for the opportunity to review the present manuscript exploring predictors of suicidal ideation and self-harm in a large representative sample of adults. The study had several noticeable strengths including a representative sampling design and use of

validated measures. I provide my comments below in hopes of clarifying and strengthening the present manuscript.

Response: We would like to thank the reviewer for recognising the strengths of this work and providing constructive feedback that has indeed strengthened the manuscript.

Detailed Comments:

1. Abstract: The authors note that model 1 excluded mental health variables, yet a significant predictor is comorbid physical and mental ill-health. I presume this is in reference to the variable "physical health conditions" which includes coding for mental health conditions. I would remove this variable for model 1 and include only in model 2 for clarity.

Response: We agree with reviewer 1 that this variable should be treated as a mental health indicator and so excluded from model 1 and instead entered into model 2. The Abstract has been revised to reflect this and Table 2 has been updated. As a result of the new model, some new effects emerged. Specifically, financial situation, non-employment, medication side effects, empathy, and hopelessness are significant predictors of suicidal ideation in Model 1. The discussion has also been amended to reflect these findings.

2. Abstract and Methods: The authors group alcohol and smoking as lifestyle factors. SUDs could also be classified as mental health disorders. The authors should provide a rationale for including alcohol and smoking as lifestyle factors rather than as a mental health variable and or include these variables only in Model 2 as mental health variables.

Response: Whilst we acknowledge that drinking and/or smoking heavily can be lifestyle responses to the symptoms of mental distress, we do not feel confident enough that this is the case for all or the majority of participants in this community sample to justify treating them as indicators of mental distress. We have therefore retained 'lifestyle factors' as the preferred terminology to describe participant responses to questions relating to smoking and alcohol behaviours within this community sample and retained them in Model 1.

3. Background: It would be helpful if the authors provide more context and background regarding the novel determinants of SI. Specifically, what is the rationale for exploring housing quality, caring responsibility, specific health conditions as predictors of SI?

Response: We have now strengthened the rationale for exploring housing quality, caring responsibility and specific health conditions as predictors of SI. (as highlighted on page 6)

4. Methods: For the creation of the SI variable, I assume the authors coded from the PHQ-9, 0 as being absent of SI and 1-3 indicating presence of SI? It would be helpful if the authors provide frequencies of people who scored 0,1,2,3 on the item 9 for detailed reporting purposes.

Response: Reviewer 1's assumption is correct. Descriptive statistics for all response options of the suicidal ideation item are now provided on page 11.

5. Methods: Related to the above comment, if there is an adequate sample size, conducting a multinomial logistic regression (0= No SI, 1 = several days, more than half the days and 2 = nearly every day) may be highly informative. People who experience SIs nearly every day are

qualitatively different than those who experience SIs several days or more than half the days and may have unique risk and protective factors.

Response: We agree that multinomial logistic regression could potentially add valuable nuance to the present findings. However, we believe the disadvantages of this technique outweigh the advantages. First, a binary logistic regression was conducted due to the limited number of people in some categories and because dichotomisation is recommended for severely skewed variables, as indicated by the Shapiro Wilk statistic. Non-dichotomisation would have resulted in very few people in many of the crosstab cells related to health conditions, making findings unreliable and conclusions difficult to draw, particularly from non-significant effects. Related to this point, a concern is that multinomial logistic regression would produce a very large and complicated output comprising 186 effect sizes. This would make interpretation and discussion more complex if effects were different in different suicidal ideation categories, potentially diluting the key messages of the paper. Second, the paper is designed to provide a broad picture of the factors that may be linked to suicidal ideation in order to stimulate more targeted follow-up research. Dichotomisation of dependent variables is very common in public health research and can provide important insights into the factors that lead to one outcome rather than the other. Indeed, it could be argued that it is important to understand the determinants of any frequency of suicidal ideation as a single thought can have the same impact on self-harm behaviours as frequent thoughts.

6. Methods: One important question I have is the calculation of the PHQ-Score. Given the authors used item 9 as their dependent measure, was this taken into account when calculating participants total score on the PHQ-9? If not, this may violate assumptions of independence and also contribute to the strong association between depression and SI.

Response: Item 9 of the PHQ-9 was excluded from the calculation of total depression scores – effectively the total PHQ score reflects the total of responses for 8 and not 9 items. This is now clearly described on page 10.

7. Method: Were the measures reported herein all measures included in the survey? If not, the authors should explicitly state so and provide a rationale for selectively choosing their measures, especially in light of their aim of identifying novel predictors.

Response: The study comprised a subset of the overall measures included in the survey the choice of which were informed by current suicide ideation theories, prior research evidence and were verified by lived experience public advisors within the project team. The methods section has been updated to reflect this. (page 9 and page 10)

8. Method: Please provide results of Little's MCAR. If the data is missing at random, it would help justify the authors choice of using list wise deletion.

Response: The Little's MCAR test is now reported on page 12, along with justification for using listwise deletion.

9. Method: I am curious from an ethics and safety perspective, if any safety checks were in place for high-risk participants, specifically during who reported experiencing SIs nearly every day.

Response: Administration of the Household Health Survey was conducted by BMG Research, an independent social research company who work to the Market Research Society (MRS) code of conduct. During data collection, participants were handed the portable tablet showing the questionnaire so that they could complete sensitive questions on their own and ensuring that the

interviewer was blind to their responses. BMG's interviewers are trained to deal with overt distress in participants and all participants were provided with details of relevant support contacts for all sensitive questions, including those relating to suicidal ideation.

10. Patient and Public Involvement: I would suggest moving the first paragraph of this section to the Methods, and incorporating the second paragraph in the discussion to help with the structure and flow of the manuscript.

Response: We agree with the Reviewers' suggestion and have moved the Patient and Public Involvement segment to the methods section and reworded it to aid clarity in line with Reviewer 2 comments detailed below. (page 9)

11. Discussion: I believe the manuscript would be strengthened by a more detailed implication section in the discussion section. More specifically, in the abstract, the authors state that interventions should focus on reduction depression and enhancing self-esteem, social, capital and empathy. How exactly could this be done, especially in a community-based context?

Response: We agree with this critique and have added an additional section highlighting the clinical implications of the findings from this study. (pages 25-26)

Reviewer: 2

Interesting paper with some important and novel results of associations between social, economic, and lifestyle factors and SI in a community sample. It is a shame there was no measure of acts of suicidal behaviour as the ideation to behaviour conversion is typically low, and therefore it is difficult to say how relevant the result may be in clinical terms or for suicide prevention efforts. The methods and analysis are appropriate, but there are some concerns around the way the content is presented as it lacks a clear focus, with some details of the study not entirely clear until well into the paper and/or a reliance on supplementary materials. Therefore, while this paper is a good candidate for publication I would recommend a thorough review, with additional clarifications, details, and a clearer focus on what exactly is novel about this work, what gap in the current literature it is addressing, and how the results might be relevant clinically/practically. Simplification of the writing would also reduce word-count and allow for some things that have been shunted to Supplementary files to be brought back into the main text. The core of the paper is however solid, and I hope the following comments will be of use to the authors.

Response: We would like to thank the Reviewer for acknowledging the strengths of this work and the value of its novel findings. Reviewer 2's extensive review of this paper and constructive feedback is greatly appreciated. We have sought to act upon this feedback and believe that the manuscript has been strengthened as a result. We agree with the Reviewer that the exclusion of measures of acts of suicidal behaviour is a limitation within this version of the survey and consistent with this view a measure of suicidal acts was included in the follow-up survey which we hope to examine in subsequent analyses. We acknowledge the formatting critiques highlighted by the Reviewer and included a section to describe the clinical implications of these findings.

Detailed comments on each section are provided below:

12. Abstract: Background. The background section within the abstract doesn't really set up the reason why this work is necessary or what questions it answers. The novelty of the study is not important here, instead, the aim of the work should be stated more clearly - what are you trying to address?

Response: We agree with the Reviewer comments and the abstract formatting comments provided by the Editor and have amended the text to more clearly identify what this study set out to address. (page 2)

13. Methods. I would like to know a bit more about how suicidal thoughts were assessed within the survey. What type of measures/response options were included. If this study is a sub analysis of a larger study, please include the name of the study and clarify the situation e.g. that the data were primarily collected for another purpose. What sort of regression analysis? Multivariable? What was controlled for?

Response: Text describing suicidal ideation measurement, the larger survey context and the variables controlled for in the logistic regression has been added to the abstract. (page 2) Further detail on suicidal ideation measurement has also been added in the main text (page 10) in response to Point 20 below.

14. Abstract. Results. What does additional statistically significant predictors were noted mean? If they weren't relevant to your conclusions they don't need to be mentioned in the results of the abstract. The model 1 and 2 (or adjusted and unadjusted models) should be referred to in the methods if necessary rather than here.

Response: "Additional statistically significant..." sentence has been removed. We have retained reference to both models in the abstract as we believe the varying results are sufficiently important to state here. (page 2)

15. Abstract. Conclusion. The conclusion is a bit abrupt and could have a more reader friendly sentence summarising the implications of the main results before going on to suggest interventions or how the results could be useful in further research or practice.

Response: This has been slightly expanded and now identifies target groups for potential interventions. (page 2)

16. Strengths and limitations: Point one should be rephrased to focus on the novelty as a strength rather than just stating that it is novel. What exactly is the novel aspect - the comment is too broad to be clear, and certainly many of the variables are things that are well known within the literature. It would be better for the paper to draw out exactly what aspects are novel right from the start.

Response: Point 1 has been redrafted to reflect the comments above. (page 3)

17. Strengths and limitations: Point five is not clear without further details, furthermore, while the overall sample is fairly large, the number with SI is not so large, so this could be a bit misleading.

Response: Point 5 has been redrafted to reflect the comments above. (page 3)

18. Background: General comments. Each statement should be supported by a reference. The background/introduction doesn't clearly demonstrate what this work is addressing that hasn't been addressed previously. The background needs to lead the reader to the conclusion that this work is necessary to address a gap in the current literature. I would caution against grouping suicidal ideation with suicide attempts and complete suicide in previous research as the ideation to action pathway is not that strong e.g. while most people who attempt or die by

suicide will have had suicidal ideation the majority of people with suicidal ideation never go on to act on these thoughts - see for example work on the ideation-to-action framework e.g. Klonsky & May 2014, 2016 etc. More focus on previous work specifically on suicidal ideation, what remains unknown and how this work addresses it would improve the paper.

Response: We agree with these comments and have redrafted the background section to focus solely upon suicidal ideation, the dependant variable within our study. We have strengthened the literature review pertinent to our investigations and have described current suicide ideation theories in support of the study rationale and the IVs selected. (pages 4-7).

19. Methods: General comments. As there has been work published already from this survey that has details of the methods, you could name the study up front and refer the reader to the paper or report for further details of the main study to reduce the amount you have to cover here. Then you can focus on the SI relevant parts.

Response: We thank the Reviewer for this suggestion and we have now referred to the recently published paper that provides a detailed overview of the methods pertinent to this survey. (page 9)

20. Methods. Measures. More details are needed in the measures section of the main text, particularly around the measures that were used to assess suicidal ideation. Rather than just adding supplementary files it would be good to have some kind of summary in the actual text and refer to the supplementary files for more detailed information on the other variables (though I'm not sure the level of detail in the supplementary files is needed either - especially if you can refer back to an existing publication or the the original references for the measures). Also, I couldn't see the suicide ideation variable in Table S1 but as the item is scores on a 4-point scale it would be appropriate to include how it was reduced to a Y/N variable. I'm sure you could combine the study measures S file and Table S1 somehow to reduce the amount of additional files.

Response: A new section has been added on 'Measures' with a summary of each measure used in the study. (highlighted on page 10)

21. Methods. Data analysis. The issue of broad dichotomisation should be picked up in the limitations section - what might be the impact of this on the results? Small point - tables usually go at the end of the manuscript at submission rather than in the text.

Response: Please see justification in point 5 (reviewer 1 section) above. The dichotomisation is now also discussed in the limitations section. (page 25).

22. Methods. Results: General comments. You could put some information about the sample demographics here instead of in the methods section. I'd like to know if there were any demographic differences in the SI and non-SI groups with the n% rather than just the ORs - numbers may be small in some analyses, but it is impossible to judge this at the moment. The tables need reviewing and simplifying - I can see why you put a shorter version in the text but this is missing information about the different levels or categories of the measures. Perhaps it would be better to include fewer variables, but in more detail (e.g. what variables are relevant, rather than just all that are significant), and remove these from the supplementary table to reduce duplication.

Response: We thank reviewer 2 for these suggestions. We have chosen to retain sample demographics in the "participants" section while descriptive statistics for all demographic variables can be found in the supplementary materials. We agree that it would be useful for readers to know

whether there were differences in the proportions of people reporting suicidal ideation dependent on demographic characteristics. This information is now provided for all variables in the descriptive statistics table in the supplementary materials (Supplementary Table 2). Finally, as suggested by the reviewer, journal format restrictions meant we could not include the full output table in the main text. However, we believe that including significant effects is more useful than including the most relevant effects. This is because researchers from different fields may disagree on which variables are most relevant or important. Therefore, we have not adjusted the in-text tables.

23. Methods. Model 2. Don't need to repeat what was adjusted for this should be covered in methods.

Response: Duplication of model descriptions has been removed.

24. PPI: A bit more detail on how exactly they contributed to the study would be good. Did their influence change any questions or procedures, how did they help interpret the findings? Is this specific to this study or are you talking about the main survey work? Page 11 line 6 - I find this sentence about the authorship a bit strange. It implies more than one person had input into the manuscript but only one person was added as a co-author. Please define further why the contribution of one person, and not the rest of the public advisors, was considered significant enough to warrant authorship. If these additional details are provided, the paragraph starting line 11 (Page 11) is not needed.

Response: The PPI section has been reworded to reflect these comments reflecting the lack of clarity in the original MS. Reviewer 2 should note that the PPI section has now been moved to the methods section in line with Reviewer 1 comments detailed earlier. (see page 9)

25. Discussion: General comments. You need to identify what specifically is the strong point of this work and emphasise that rather than just repeating that a lot of variables were included. If the info around housing, lifestyle and wellbeing is novel - concentrate on that. The community sample is also a bonus as a lot of work in suicidology is clinical in nature. The principal findings are a bit long. Rather than just repeating what is significant as in your results section, you could focus on the most relevant or novel results. The structure of the discussion could be tightened up to make it easier for the reader to grasp the main points of the work. There is a lack of interpretation and translation of the results and an overemphasis on whether the results are consistent with previous research. This take up a lot of the text and as these are not new information could instead be summed up with a few simple sentences e.g. Many of the results were consistent with previous research on predictors of SI, such as X Y and Z (with references). This would free up space to focus on the novel results and include further interpretation. See detailed comments below:

Response: We agree with Reviewer 2's comments and have redrafted the Discussion section to be more focused on the key, novel results and our interpretation of them. (pages 19-21)

Clinical implications of these findings have also been included within a specific section. (pages 25-26)

26. Discussion: Page 11, line 40-46. This sentence is not a result and is repeating what we already know, would recommend removing.

Response: This sentence has been removed.

27. Discussion: Page 11, paragraph starting line 49. This largely re-states what we have already been told in the results section. It would be better to concentrate on those results that

have some kind of clinical or theoretical relevance - what results are novel, and how should these be interpreted?

Response: We agree with this comment and the paragraph has been removed.

28. Discussion: Please be careful of extraneous words E.g. the sentence “Unsurprisingly, depression was the most important risk factor, whereby those suffering from depression were seven times more likely to report SI than those without” could just read something like “Those suffering from depression were seven times more likely to report SI than those without.” In fact, this could be combined with the later sentence (Page 12, line 29-31) about depression, anxiety and paranoia being known factors in current literature (a general writing tip - see how many words you can remove form a sentence and still have it make sense - I guarantee it is less than you think!).

Response: We agree with this comment and the wording has been amended accordingly. (page 19)

29. Limitations. The limitations section is seriously lacking. Issues that could be covered - geographical location of the survey and how this impacts generalisability to other areas. Only adults were included - self-harm is common in young people, what about SI? No acts of suicidal behaviour were recorded, and there was no follow-up to establish if thoughts were acted on - could these people be the ones who never act on these thoughts - if so, how useful is this data? The issues around collapsing categories could be raised here e.g. BAME people were all placed in the same group, but there might be nuances and cultural aspects that this hides. What about the 1-4 levels of SI? The reader does not get to see these at all (actually that is something you could include in the results - just n% of the SI group) and it could be that these are all low-level thoughts.

Response: We accept these additional limitations should be noted and they have been added to an expanded limitations section. Descriptive data on the four levels of SI have been added to the manuscript (see comment 4 from Reviewer 1, above).

30. Conclusion: The conclusion starts strong, with some interpretation that I would like to see more of in the rest of the discussion. But it ends with an overstating of the relevance of this work - see comments on additional limitations to consider. More detail needs to be added as to how this work can inform future research, clinical management, and reduction and prevention strategies. Again, you could focus on the more novel aspects of your work. What about public health initiatives to tackle community SI? BAME or LGBT specific interventions?

Response: We have acted upon this constructive feedback by including additional interpretation within the discussion section (pages 19-21) and an additional section detailing the clinical implications of this work (pages 25-26). The conclusions section has been edited to highlighting the importance of identifying risk and protective factors pertinent to suicidal ideation and public health initiatives that could be informed by the findings from this study. (page 26)

VERSION 2 – REVIEW

REVIEWER	Hyoun S. Kim Ryerson University, Canada
REVIEW RETURNED	22-Oct-2020
GENERAL COMMENTS	I would like to thank the authors for their careful attention paid to my comments in the first review. I feel the authors have addressed all my concerns. Furthermore, I appreciate the authors justification

	with including alcohol/smoking as lifestyle factors and justification for not considering a multinomial logistic regression. I wish the authors the best with their work.
--	---